# The Impact of the COVID-19 Pandemic and Lockdown on Macular Hole Surgery Provision and Surgical Outcomes: A Single-Centre Experience

**DOI:** 10.3390/jcm11133678

**Published:** 2022-06-26

**Authors:** Georgios D. Panos, Olyvia Poyser, Humera Sarwar, Dharmalingam Kumudhan, Gavin Orr, Anwar Zaman, Craig Wilde

**Affiliations:** Department of Ophthalmology, Queen’s Medical Centre Campus, Nottingham University Hospitals, Nottingham NG7 2UH, UK; olyviapoyser@hotmail.com (O.P.); humera.sarwar@nhs.net (H.S.); dharmalingam.kumudhan@nuh.nhs.uk (D.K.); gavin.orr@nuh.nhs.uk (G.O.); anwar.zaman@nuh.nhs.uk (A.Z.); craig.wilde@nuh.nhs.uk (C.W.)

**Keywords:** macular hole surgery, COVID-19, pandemic, lockdown, policies, retinal surgery

## Abstract

Purpose: We aimed to report the impact of the COVID-19 pandemic and related health policies and restrictions on the provision and efficacy of macular hole (MH) surgery. Methods: We carried out a retrospective cohort study. Two MH patient cohorts, those treated during the COVID-19 pandemic (12 months) and the pre-COVID-19 period (12 months before the lockdown) were reviewed and compared. Patient characteristics, time to consultation and surgery, MH size, baseline and postoperative visual acuity (VA) and failure rate were recorded and analysed. Results: A reduction of 43% in MH surgery occurred during the COVID-19 period (93 eyes vs. 53 eyes). Mean time to consultation and time to surgery increased significantly (52.7 days vs. 86.3 days, *p* < 0.01 and 51.3 days vs. 83.6 days, *p* = 0.01, respectively), while mean baseline and postoperative vision was significantly lower in the COVID-19 group (0.75 LogMAR vs. 0.63 LogMAR, *p* < 0.01 and 0.61 LogMAR vs. 0.44 LogMAR, *p* < 0.01, respectively). The median MH size was significantly larger in the COVID-19 group (296 μm vs. 365 μm, *p* = 0.016), and the failure rate increased from 7.6% to 15.4% (odds ratio 2.2 (95% CI: 0.72–6.8)). Conclusions: Our findings suggest the COVID-19 pandemic caused a significant reduction in MH surgery, increased waiting times and led to poorer surgical outcomes. For future pandemics, better strategies are required that allow semi-elective and elective surgery to continue in a timely fashion. Health providers should preserve the delivery of ophthalmological care, with enhanced encouragement to seek medical help for acute symptoms.

## 1. Introduction

Since coronavirus disease 2019 (COVID-19) was declared a pandemic by the World Health Organization (WHO), many countries, including the United Kingdom (UK), have imposed national lockdowns. The related restrictions significantly affected healthcare services and the provision of patient care due to cancellations of routine outpatient appointments and surgery. In order to minimise the risk of COVID-19 transmission, patients were discouraged to attend hospital visits, unless critically necessary.

A study from Moorfield’s Eye Hospital, London, UK, reported a significant reduction in clinical and surgical activities, including in the A&E department (>50% reduction), the medical retinal service (did not attend (DNA) rate increased to 24.9%) and vitreoretinal (VR) service (average drop of 62% in retinal detachment (RD) surgery) [1]. Audits from our department demonstrate during the first UK lockdown (late March–June 2020), outpatient and surgical activities were reduced by 63% and 67%, respectively, compared to the same period the previous year. This resulted in increased breaches of the 18-week referral to treatment (RTT) target in outpatient appointments (by 75%) and surgeries (by 471%) [2]. Moreover, during this period, patients with RD presented with a longer duration of symptoms, with the proportion of macula-off detachments and proliferative vitreoretinopathy (PVR) increasing, resulting in an escalated risk of worse outcomes in surgery [3]. 

This study reports the effect of the COVID-19-related policies and restrictions on the efficacy of the macular hole (MH) surgery at Queen’s Medical Centre (QMC), which is one of the largest hospitals in the UK. 

## 2. Materials and Methods

This study was performed in accordance with institutional guidelines and the ethical standards of the Declaration of Helsinki. The institutional review board approved the protocol (reference no. 21-219C).

This is a single-centre, retrospective cohort study. Consecutive patients who underwent MH surgery during the first year of the COVID-19 pandemic (April 2020–March 2021) were included in the study (study/COVID-19 group), while patients who underwent MH surgery one year before the pandemic (April 2019–March 2020) were included to serve as the control group (pre-COVID-19 group). Only the first eye of each patient was included in the case both eyes underwent surgery, either in the COVID-19 or pre-COVID-19 period, in order not to violate the statistical assumption of independence of observations. Eyes which developed postoperative complications not related to the delay of the surgery (i.e., RD, choroidal neovascularisation) were excluded from analysis. 

The study protocol included collection of demographic data including age, gender, recorded **symptom duration** at clinical presentation, **time to consultation** in calendar days from day of referral (either from eye casualty, or optician or general practitioner (GP) or other hospital/clinic) to the VR clinic appointment, **time to surgery** in calendar days from the day of listing in the VR clinic, **lens status** (phakic, aphakic or combined in the case patient had combined surgery), **best corrected visual acuity (BCVA)** in LogMAR units at baseline and 3 months postsurgery, MH **size** and **need for further MH surgery** in the case of failure/nonclosure. 

### 2.1. Visual Acuity

BCVA was recorded on LogMAR charts with a score of 0.02 equal to 1 letter. Postoperative BCVA was recorded at 3 months since this period is sufficient for anatomical and functional recovery following MH surgery and gas dispersion, while the effect of cataract formation in phakic eyes is minimal. 

### 2.2. Macular Hole Size

Macular hole size was measured manually on the optical coherence tomography (OCT) with the Heidelberg Spectralis software caliper as a line drawn roughly parallel to the retinal pigment epithelium, at the narrowest distance between the hole edges.

### 2.3. Time to Consultation and Time to Surgery

Any delays not related to the pandemic or to normal waiting, such as cancellations/re-scheduling by the patient, incorrect booking or other unforeseen circumstances, were taken into account to reflect the real waiting time.

### 2.4. Surgical Technique

The same experienced consultant vitreoretinal surgeons (GO, AZ, DK, CW) operated on the patients of both groups. All patients underwent pars plana vitrectomy (core and peripheral) with three 23-gauge ports. Posterior vitreous detachment was induced, unless it was present. The internal limiting membrane (ILM) was stained with MembraneBlue Dual (DORC, Zuidland, The Netherlands) and peeled with end-gripping forceps. Following internal 360-degree search with indentation, air–fluid exchange (AFX) was performed with Charles flute followed by air–gas exchange. For all MH patients, perfluoropropane (C_3_F_8_) 12% (Alchimia, Ponte San Nicolò, Italy) was used as tamponade. Patients were advised to lie facing down for 10 days and attended regular follow-up appointments (week 1, week 3, week 6 and month 3) for the first 3 months after surgery. 

### 2.5. Statistical Analysis

Statistical analysis was performed with MedCalc Statistical Software, version 18.2 (MedCalc Software bvba, Ostend, Belgium; http://www.medcalc.org; accessed on 19 March 2022). Data distribution was checked with the Shapiro–Wilk test and Q–Q plots. Parametric and non-parametric tests were applied accordingly. Data are presented as mean ± standard deviation or median (range). Values of *p* less than 0.05 were considered statistically significant.

## 3. Results

A total of fifty-three (53) eyes of 52 patients underwent MH surgery during the COVID-19 period, whereas 93 eyes of 91 patients underwent MH surgery during the pre-COVID-19 period, suggesting a reduction of 43% due to the lockdown secondary to the pandemic. A total of 52 (52) eyes of 52 patients in the COVID-19 period and 79 eyes of 79 patients in the pre-COVID-19 period met the inclusion criteria and were included in the analysis as the study and control group, respectively. The two groups were similar in terms of age, sex ratio and lens status; however, the mean symptom duration in the study group was longer by 1.5 months compared to the control group (5 months ± 3.4 vs. 3.5 months ± 2.0, *p* = 0.01). Demographic characteristics of both groups are depicted in Table 1. 

### 3.1. Time to Consultation

Mean time to consultation in the study group was significantly longer compared to the control group (86.3 days ± 71.7 vs. 52.7 days ± 25.7; Welch test, *p* < 0.01, Figure 1A), suggesting a significantly increased waiting time from the day of the referral to the VR clinic appointment.

### 3.2. Time to Surgery

Similarly, the mean waiting time to surgery from the day of listing at the VR clinic appointment increased by approximately a month during the pandemic, (83.6 days ± 90.1 vs. 51.3 days ± 22.5; Welch test, *p* = 0.01, Figure 1B).

### 3.3. Visual Acuity

Mean baseline BCVA in the study group was 0.75 LogMAR ± 0.27, whereas the baseline BCVA in the control group was 0.63 LogMAR ± 0.20; this difference was statistically significant (Welch test, *p* < 0.01, Figure 1C). Similarly, mean postoperative BCVA in the study group was significantly worse compared to the postoperative BCVA of the control group (0.61 LogMAR ± 0.31 vs. 0.44 LogMAR ± 0.23; Welch test, *p* < 0.01, Figure 1D). However, the difference in the BCVA improvement between the groups was not statistically significant (0.138 LogMAR ± 0.29 vs. 0.185 LogMAR ± 0.24; t-test, *p* = 0.33, Figure 1E). 

An improvement of at least 2 lines on the LogMAR chart was observed in 47.05% of the patients in the study group, and an improvement of at least 3 lines was observed in 31.37% of the study group patients following MH surgery. For the patients in the control group, the same proportions were 52.6% and 38.15%, respectively (Figure 2A,B). 

### 3.4. Macular Hole Size

Median MH size in the study group was 365 μm (range: 182–1283) while in the control group the median MH size was 296 μm (range: 50–829); this difference was found to be statistically significant (Mann–Whitney U test, *p* = 0.016, Figure 1F).

### 3.5. Failure Rate

Eight eyes (15.4%) in the study group required further surgery due to non-closure of the MH during the first attempt, whereas the MH surgery failed at the first attempt in the control group in only six eyes (7.6%) (odds ratio 2.2 (95% CI: 0.72–6.8)).

## 4. Discussion

The COVID-19 pandemic has greatly impacted patients and healthcare workers, both directly and indirectly. It created challenges for the NHS, resulting in a sharp decline in referrals to secondary care and a drop in hospital activity. We investigated whether these changes altered surgical outcomes of MH patients during the COVID-19 pandemic. We found a significant reduction in MH surgery during the COVID-19 pandemic with an associated suggestion of delayed presentation and increased waiting times from the day of referral to the day of the clinical and surgical appointments. These delays appear cumulative to such an extent that an increased mean MH size was noted among patients presenting during the pandemic. These changes are known to pose a significant risk for worse functional outcomes, and we demonstrated worse vision at presentation during the pandemic, with worse surgical outcomes postoperatively. Failure of anatomical closure was twice as high among patients operated on during the pandemic, compared to those who were treated the preceding year. 

For the purposes of this study, we considered the first year of the pandemic as the COVID-19 period, commencing in April 2020 at the start of the national lockdown in the UK. During this year, wider social restrictions and hospital policies varied, depending on the epidemiological status of the country. Periods of strict full lockdown and significantly reduced clinical and surgical activity were followed by periods of partial lockdown or near normality with only slight reductions in clinical and surgical activity. We defined the pre-COVID-19 period as the same period of the previous year (April 2019–March 2020).

Although MH surgery is not considered an emergency, it is well-documented that successful outcomes and the level of visual recovery are affected by the duration of the MH [4,5]. A recent study by the BEAVRS macular hole outcome group demonstrated that an MH duration longer than 4 months reduces the chance of visual success by 50% [6]. During the first year of the pandemic, patients presented in our VR clinic with longer symptom durations. The reasons for this are complex and varied, but are likely secondary to patients being reluctant to attend (or failure to gain adequate or prompt access to) eye casualty or GP/optometry appointments, owing to the perceived risk of coronavirus infection. Hospitals treating COVID-19 patients were perhaps considered high-risk environments and avoided by some, particularly the predominantly elderly ophthalmic populations. During the national lockdown, where the message was to ‘stay at home, protect the NHS, save lives’, many patients may have deferred seeking help for other health conditions, particularly ones that were considered minor. Intense media coverage of the pandemic, frequently reporting worrying stories, such as shortages of personal protective equipment (PPE) and hospital capacity issues, may have confounded the issue. Healthcare workers and hospitals were possibly substandard in reassuring patients to attend for non-COVID-19-related disease. Significant delays in clinical appointments were caused by reduced non-urgent clinical activity that followed the NHS England and Royal College of Ophthalmologists Directives, in order to tackle the spread of the virus. Outpatient clinic capacity was reduced to allow enhanced cleaning and to comply with social distancing rules within clinical waiting areas. Time to treatment was further delayed secondary to reduced theatre capacity. Operating lists were cancelled, and lists had to be shared across ophthalmic sub-specialties, secondary to redeployment of doctors (particularly anaesthetists and trainees) and nursing staff to medical wards, intensive care units or dedicated COVID-19 wards. Staff sickness from COVID-19 was high, and staff absences secondary to shielding among those with COVID-19-positive contacts or those deemed high risk if they were to suffer from COVID-19 infection also contributed. Clinicians often had no influence on top-down policies instituted by the government and enacted through senior hospital management without debate or discussion. Pathways were changed and staff redeployed in ways that ultimately prioritised COVID-19, but significantly compromised other patient cohorts. The cumulative effect was the development of larger MHs, worse baseline and postoperative visual acuity and increased failure rate (over two times) in patients who were referred, seen in clinic, listed for surgery and operated on for MH during the first year of the pandemic. 

There have been numerous studies investigating the effect of the first 3 months of the lockdown period on ophthalmic surgeries and consultations [3,7,8,9,10], most of which focus on retinal detachment surgery [3,9,10]. To the best of our knowledge, this is the first study to report the longer-term impact of COVID-19-related policies and lockdown restrictions on MH surgery.

In a recent study, Awan et al. reported a 45.5% reduction in retinal surgeries during the first lockdown in Pakistan (March–June 2020) [7]. During this period, no MH surgery was performed. A retrospective analysis from six ophthalmology departments in Italy reported a reduction of 77% in MH surgery during the first month of the national lockdown (10 March–9 April 2020) compared to the month before (43 cases vs. 10 cases) [8]. Awad et al. reported a statistically significant increase in retinal detachment cases presenting with PVR during the first 2 months of the lockdown period in Nottingham, UK (March–May 2020) [3], while in Scotland, the weekly mean number of rhegmatogenous retinal detachments reduced from 18.2 before lockdown to 8.6 during the first 5 weeks of lockdown, decreasing the annual incidence from 17.37/100,000/year to 8.21. Simultaneously, macula-off retinal detachments increased at presentation by 10% [9]. Similarly, a study from Beijing, China, reported a 55.9% reduction in retinal detachment surgery during the COVID-19 pandemic but showed no significant difference in the severity of PVR and in surgery outcomes [10]. 

## 5. Conclusions

In summary, we report a cohort of MH patients treated during the first year of the COVID-19 pandemic. Significant disruption to healthcare service provision caused a reduction in MH surgery and longer waiting times, leading to poorer anatomical and functional outcomes. For future pandemics, better strategies are required that allow semi-elective and elective surgery to continue in a timely fashion. Health providers should preserve the delivery of ophthalmological care, with enhanced encouragement to seek medical help for acute symptoms. Massive investment within the NHS with additional resources will be required. 

## Figures and Tables

**Figure 1 jcm-11-03678-f001:**
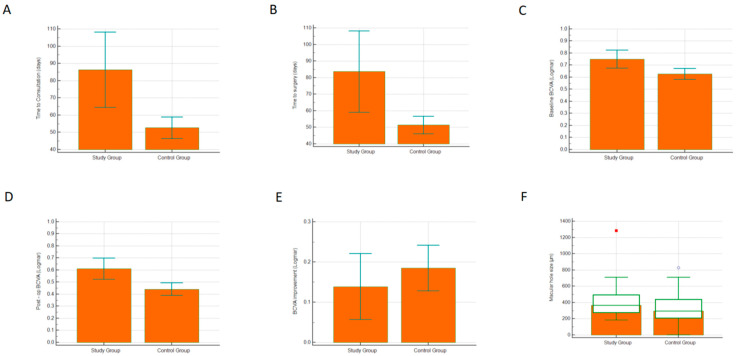
(**A**) Time to consultation (days), (**B**) time to surgery (days), (**C**) baseline best corrected visual acuity (LogMAR units), (**D**) post-op best corrected visual acuity (LogMAR units), (**E**) best corrected visual acuity improvement (LogMAR units), (**F**) macular hole size (μm).

**Figure 2 jcm-11-03678-f002:**
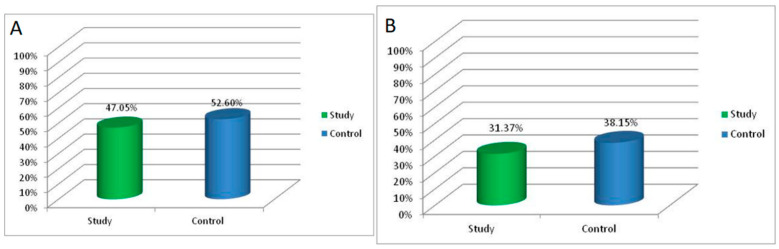
(**A**) Proportion of patients with an improvement of at least 2 lines on the LogMAR chart (**B**). Proportion of patients with an improvement of at least 3 lines on the LogMAR chart.

**Table 1 jcm-11-03678-t001:** Demographic characteristics of the groups.

	Control Group (Pre-COVID-19)	Study Group (COVID-19)	*p* Value
Total number of eyes/patients operated on	93/91	53/52	
Number of eyes/patients included	79/79	52/52	
Age (years, mean ± SD)	67.6 ± 7.3	68.6 ± 6.6	0.48
M:F	28:51 (35.4% M, 64.6% F)	16:36 (30.8% M, 69.2% F)	0.57
Lens status	64.9% phakic	60.4% phakic	0.60
	18.2% pseudophakic	18.8% pseudophakic	0.93
	16.9% combined surgery	20.8% combined	0.57
Symptom duration (at VR clinic appointment)	Median: 3 months (1–8)	Median: 4 months (1–12)	0.01
	Mean: 3.5 months ± 2.0	Mean: 5 months ± 3.4	

## Data Availability

Data are available from the authors upon reasonable request.

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
