# Peer review of "The Impact of the COVID-19 Pandemic and Lockdown on Macular Hole Surgery Provision and Surgical Outcomes: A Single-Centre Experience"

_jcm, 2022, doi:10.3390/jcm11133678_

Round 1

Reviewer 1 Report

The authors present essential findings collected during the recent COVID-19 healthcare system crisis. Results seem to be consistent with already collected data.
Unfortunately, this study does not present information on whether surgeries were performed by the same professional or not. Moreover, the authors do not mention what surgical technique was applied. These facts do not devalue the clinical importance of the study but could impact the evaluation of surgical outcomes.

Author Response

Dear reviewer, 

Thank you for your comments and for reviewing our manuscript. 

Patients of both groups underwent standard Pars plana vitrectomy + ILM peel + C3F12% by the same experienced consultant vitreoretinal surgeons. 

The following paragraph was added to the revised version (methods section). 

2.4. Surgical Technique  

      The same experienced consultant vitreoretinal surgeons (GO, AZ, DK, CW) operated the patients of both groups. All patients underwent pars plana vitrectomy (core and peripheral) with three 23 gauge ports. Posterior vitreous detachment was induced, unless it was present. The internal limiting membrane (ILM) was stained with MembraneBlue Dual (DORC, Zuidland, the Netherlands) and peeled with end gripping forceps. Following internal 360 degree search with indentation, air – fluid exchange (AFX) was performed with Charles flute followed by air – gas exchange. For all MH patients, perfluoropropane (C3F8) 12% (Alchimia, Ponte San Nicolò, Italy) was used as tamponade. Patients were advised to posture face down for 10 days and attended regular follow – up appointments (week 1, week 3, week 6 and month 3) for the first three months after surgery.       

Thank you again for your time and efforts 

Best wishes 

Reviewer 2 Report

This research addresses the question at what extent did the pandemic slow-down ophthalmic care, and what were the consequences. The topic is not original, but the studied sub-specialty is: I am not aware of similar studies involving the retinal pathology. Therefore the paper is interesting as it underlines that non-emergency conditions also suffered from the pandemic. The paper is well-written, and easy to understand.  This is of high social value. It will contribute to better appreciation of ophthalmology in medicine.The conclusions are consistent with the purpose and with the results, and they added the main question posed.

(Please correct "logmar" into "LogMAR" which is the usual notation for visual acuity.)

Author Response

Dear reviewer, 

Thank you for your comments and for reviewing our manuscript. 

We changed "logmar" into "LogMAR" (which is indeed the correct form) throughout the manuscript (changes highlighted).  

Thanks again for your time and effort. 

Best wishes